# The Neurological Basis of Developmental Dyslexia and Related Disorders: A Reappraisal of the Temporal Hypothesis, Twenty Years on

**DOI:** 10.3390/brainsci11060708

**Published:** 2021-05-27

**Authors:** Michel Habib

**Affiliations:** Cognitive Neuroscience Laboratory, Neurodys Institute, Aix-Marseille University, UMR 7291 Marseille, France; michel.habib@univ-amu.fr

**Keywords:** brain imaging, dyslexia, learning disorders, music, phonology, rhythm, time processing, tractography

## Abstract

In a now-classic article published a couple of decades ago (Brain, 2000; 123: 2373–2399), I proposed an “extended temporal processing deficit hypothesis of dyslexia”, suggesting that a deficit in temporal processing could explain not only language-related peculiarities usually noticed in dyslexic children, but also a wider range of symptoms related to impaired processing of time in general. In the present review paper, I will revisit this “historical” hypothesis both in the light of a new clinical perspective, including the central yet poorly explained notion of comorbidity, and also taking a new look at the most recent experimental work, mainly focusing on brain imaging data. First, consistent with daily clinical practice, I propose to distinguish three groups of children who fail to learn to read, of fairly equal occurrence, who share the same initial presentation (difficulty in mastering the rules of grapheme–phoneme correspondence) but with differing associated signs and/or comorbid conditions (language disorders in the first group, attentional deficits in the second one, and motor coordination problems in the last one), thus suggesting, at least in part, potentially different triggering mechanisms. It is then suggested, in the light of brain imaging information available to date, that the three main clinical presentations/associations of cognitive impairments that compromise reading skills acquisition correspond to three distinct patterns of miswiring or “disconnectivity” in specific brain networks which have in common their involvement in the process of learning and their heavy reliance on temporal features of information processing. With reference to the classic temporal processing deficit of dyslexia and to recent evidence of an inability of the dyslexic brain to achieve adequate coupling of oscillatory brain activity to the temporal features of external events, a general model is proposed according to which a common mechanism of temporal uncoupling between various disconnected—and/or mis-wired—processors may account for distinct forms of specific learning disorders, with reading impairment being a more or less constant feature. Finally, the potential therapeutic implications of such a view are considered, with special emphasis on methods seeking to enhance cross-modal connectivity between separate brain systems, including those using rhythmic and musical training in dyslexic patients.

## 1. Introduction

Developmental dyslexia, or specific learning disorder of reading hereinafter referred to as “dyslexia”, is the most common form of specific learning disorder. The two major international classifications, DSM-5 (American Psychiatric Association, 2013) and ICD-11 (still in preparation), have relatively clear and globally similar definitions that include a number of criteria: a reading acquisition difficulty resulting in a lag compared to the performance of average individuals on standardized reading tests, a significant repercussion of this difficulty on school/academic achievement and on the use of reading in daily life, and finally the normality of intelligence and the absence of other pathologies likely to interfere with the learning process.

Over the last twenty years, advances in several fields of neuroscience and neuroimaging have refined our comprehension of dyslexia as a neurological disorder, mainly in an attempt to gain some coherence between the various existing theories. The starting point of the present paper will be the general overview described in my 2000 paper [1], where I proposed to extend the explanatory power of the “temporal processing deficit theory”, that was emerging at the time as a likely candidate to account for the clinical complexity of dyslexia and reconcile the different theoretical approaches. Since then, a considerable amount of relevant literature has accumulated, yielding significant advances just as much in the clinical, neuroimaging or experimental fields, providing new frames for a temporal processing account of dyslexia. The present review is organised into two parts following a three-step progression: (1) an outline of the disorder’s main clinical presentations, considering first the classic and most widely recognized linguistic/phonological form of dyslexia and then the two other less often acknowledged but equally important visual–attentional and dyspraxic forms of dyslexia; (2) an updated overview of the neural substrate of each of the three clinical forms and their respective comorbidities, highlighting a common feature of impaired connectivity derived from modern neuroimaging contributions; (3) a recall of the classical temporal processing theory of dyslexia and a summary of recent developments around the theme of impaired timing processes in dyslexia, with special emphasis on the topic of brain oscillations. Finally, it is reasoned that combining the disconnectivity and temporal impairment data may lead to a new overall comprehension of dyslexia and other specific learning disorders with straightforward implications at the interface between medicine and education. Ultimately, an additional goal of this article is to make the case for fostering closer ties between scientific research and clinical observations in the field of neurodevelopmental disorders, for example in considering temporally driven remediation approaches as a common means to treat various forms of learning disorders.
**Part one. Dyslexia and its comorbidities: from clinical presentation to disrupted connectivity**

## 2. Main Clinical Features

### 2.1. The “Classic” Presentation of Phonological Dyslexia

In clinical practice, the most common diagnostic situation is undoubtedly that of a child consulting in the first two years of primary school for difficulties with written language.

The errors noted are often of a phonological nature, such as voiced/unvoiced errors (p/b, t/d …) sometimes associated with symmetrical letter errors (d/b, p/q), the stake rapidly becoming the child’s ability to create an orthographic lexicon, i.e., an overall visual representation of readily accessible word forms. Depending on the degree of transparency/opaqueness of the native language (e.g., German vs. English), the reading disorder will take different forms, impaired fluency being more common in transparent languages whereas phonemic errors will predominate in opaque ones.

Quite often, this type of dyslexia is associated with impaired short-term memory, usually mainly on auditory input, including learning counting tables, sometimes as a part of real dyscalculia. In the purest forms, however, working memory, as well as long-term memory and attentional processes, are found falling within normal limits. It is also in this classic form of dyslexia that one can find either in the patient’s past history, or even on examination, evidence of some degree of oral language delay or impairment, beyond mere phonology, for example, impaired fluency of oral expression, poor vocabulary, or even weak syntactic competences. Later on, some form of pragmatic disorder often appears, characterized by the distortion of the narrative coherence of oral or written productions. The observed subsequent evolution of early findings varies considerably according to a number of factors, including the initial severity, the rapid implementation of suitable rehabilitation and school facilities, but also the child’s own resources, both in terms of general intellectual efficiency and ability to deal emotionally with difficulty. Very early on it is noted that written expression, which is intrinsically linked with the reading disorder, is flawed, not so much the writing gesture itself, which in some cases is totally unaffected, but rather in the orthographic form of the words, which are distorted by multiple errors: phonological confusions, elisions, substitution of letters, etc., generally referred to as spelling errors (dysorthographia).

### 2.2. The Non-Verbal or “Visuo-Attentional” Subtype of Dyslexia

Apart from this typical case of phonological dyslexia (which should more accurately be called “linguistic” in reference to the above-mentioned demonstration of wider language impairment), an apparently similar scenario can occur in a very different context in which no linguistic disorder is found, even in the most subtle phonological tests, whereas grapheme-to-phoneme conversion is still profoundly impaired. Here, characteristics indicative of an attentional disorder are often noted, and are confirmed by low scores on specific cognitive tests, including working memory, and in some cases, behaviours typical of an Attention Deficit Disorder with or without Hyperactivity (ADHD), such as impulsivity and/or more or less visible motor agitation. Very typically, these children are also challenged in areas other than reading, in fact in all tasks requiring sustained and/or shared attention, as in dual-task situations, such as listening attentively to the teacher while writing or calculating, for example. Although it could be argued that the general attentional deficit alone can explain the observed difficulties in learning to read, there is convincing evidence that these difficulties result from a specific impairment of particular mechanisms that assign attentional resources to the visual processing of letter strings while reading. Indeed, there is substantial agreement that these non-linguistic forms of dyslexia can be ascribed to a visual–attentional mechanism [2] (see below) deriving from impaired functioning of the bilateral temporo-parietal circuits of attention [3] clearly separated from the left hemispheric frontal–temporal circuits of language. Such faulty involvement of attention in the reading process has been further analysed as either a defective (“sluggish”) disengagement of attention during the rapid presentation of successions of letters [4] or as a reduced visual attentional span [5]. In any case, the dyslexia caused by this type of mechanism tends to take a specific form, akin to the so-called “surface dyslexia pattern” [6] where reading is very slow, hesitant, albeit with few errors, and particularly effortful, thus generating a great deal of cognitive fatigue that will potentially impair every single classroom activity. It is also in this type of dyslexia that the worst spelling problems are encountered, probably because the systematic decoding procedure does not allow the subject to construct an orthographic lexicon and hence to gain automatic access to orthographic selection and production, with significant variability according to the characteristics of the orthographic system of each individual native language [7]. Among the various tools available for testing attentional processes, those available in automatized and computerized forms have practical advantages, including specific tools that have been developed to test the presumed underlying mechanism [8], but the neuropsychologist and neurologist will usually prefer “paper and pencil” ones, which are more adaptable to each clinical situation. It is noteworthy that normal performance on these general attentional tasks does not mean normal attentional processes, especially in those dyslexic individuals with a high intellectual capacity [9].

### 2.3. A So-Called “Dyspraxic” Form of Dyslexia

Finally, in some cases, the initial reading disorder will tend to fade during the first few months and be replaced by a predominant difficulty in written expression, whether or not already identified since kindergarten in the form of a dysgraphia or a more extensive disorder in motor coordination, most generally labelled dyspraxia. Although the term “dyspraxic” is used here, this does not mean that these children fall under the DSM-5 heading of Developmental Coordination Disorder, since most cases will not meet the criteria for this diagnosis. We are simply pointing out some traits shared with dyspraxic children, for example, awkward handwriting, or difficulty copying geometrical drawings as well as visuo-spatial abilities which stand below what is expected given their age and intellectual level. Most typically, children’s reading abilities, albeit initially a serious matter of concern, will improve within the first two years to the point where they no longer need support, while, at the same time, the impact of their writing difficulties on their school productivity gradually grows, due to increasing demands for speed and accuracy in the academic system. Written productions most often associate patterns of dysgraphia and multiple spelling errors, due to combined and reciprocal interaction between the linguistic and motor aspects of the process of writing [10]. In some cases, dyslexia may remain problematic, often indicating neuro-motor involvement of the ocular apparatus, with eye-tracking problems or abnormal saccades that affect the fluidity of eye movement during the act of reading [11], as part of a more general coordination impairment. In these cases, rehabilitation by an orthoptist (or optometrist depending on each country’s care system) may significantly improve reading ability [12], although it is often unclear whether such oculo-motor problems are causally linked to the reading difficulties or only contribute to their severity [13]. On the other hand, the problem with written expression, manifested by the awkwardness of the writing gesture, irregularity of the letters’ baseline and ultimately by a gradually increasing gap between the academic requirements and the pupil’s personal potential for compensation, will often lead, more or less rapidly and exclusively, to the replacement of handwriting by typing on a keyboard.

Thus, in the latter case, it is neither the phonological system nor the attentional processes that are to blame, but rather impaired motor coordination and visuo-spatial abilities. Accordingly, a report by a psychomotor or occupational therapist will often indicate some difficulty as regards visual–spatial processes, as in the classic task of copying the Rey–Osterreith figure, where its structure is poorly perceived and incorrectly reproduced, as well as difficulties in mastering temporal notions (see below).

One subject of debate concerns the concept of dysgraphia itself, most often considered as a form of coordination disorder specifically involving handwriting movements, while some definitions seem to focus instead on the spelling disorder, with little if any reference to the gesture itself [14]. The issue is rendered complex, however, by the high incidence of association between dyslexia and dysgraphia, and the usual reciprocal worsening effect of the two conditions [15].

## 3. The Neuroanatomy of Dyslexia: A Selective Update

Over the past twenty years, the understanding and management of learning disabilities have greatly benefited from the contribution and advances in brain imaging, especially in the field of dyslexia research where most scientific endeavour has been concentrated. Accordingly, a rapid overview of the available imaging literature would easily show a disproportionately larger bulk of data on dyslexia compared to other learning disorders, and also, within dyslexia research, among various competing views. In fact, it is becoming clearer that the initial dogma of phonological disorder as the unique mechanism leading to reading impairment has generated a huge bias, focussing attention exclusively on linguistic models and driving both fundamental and applied research toward the same circular reasoning between dyslexia and phonological impairments, thereby confusing causation and correlation. It is thus important to keep in mind, throughout the following sections, that the quantity of available research on both functional and morphological anatomy in phonological dyslexia does not necessarily reflect its actual relevance in terms of occurrence and clinical importance.

### 3.1. Phonological Dyslexia: Mainly But Not Exclusively a Left-Hemisphere Problem

The brain substrate of this linguistic form of dyslexia has now been firmly established through numerous independent studies using functional brain imaging. Figure 1 summarizes the main findings obtained from these studies, pointing to the dysfunction of a left-hemisphere network including Broca’s area in the posterior–inferior frontal lobe and Geschwind’s area at the left temporo-parietal junction, both anomalies being best revealed when dyslexic individuals (children or adults) are scanned while performing phonological tasks, such as a rhyming task (e.g., say whether or not two words sound the same at the end). When subjects are asked to read words or sentences aloud, they will very frequently underactivate the same areas, but also, and probably even more clearly, another area whose specific role in reading has been one of the major revelations of research in neurophysiology over the last two decades: the visual word form area (VWFA [16]), located in the left fusiform gyrus, i.e., on the lower edge of the hemisphere, midway between the temporal and occipital poles, close to the visual cortex. This zone is considered as the one responsible for the assignation of a linguistic status to the visual stimuli represented by sequences of letters during the act of reading. It specializes in the very first moments of learning to read [17] and seems to be the most significantly underactive part of the brain in dyslexic children and adults (at least in so-called alphabetic languages). Since then, several meta-analyses [18,19] have confirmed that these three zones are consistently activated during reading and/or during oral or visual phonological tasks and that they dysfunction in dyslexics. Specific literature, that will not be surveyed in detail here, has been devoted to describing variations of this general brain activity landscape according to maternal language, contrasting deep orthographies, like English, with weak phonological/orthographic correspondence, and shallower orthographies, like Italian [20] and alphabetic vs logographic (like Chinese) writing systems [21]. The most recent developments of this literature seem to favour a universal tendency, across various orthographic and writing systems, for underactivation of these three left-hemispheric zones [22].

Besides the functional imaging data summarized in the above paragraphs, the dyslexic brain also presents structural differences or peculiarities that were demonstrated using various structural imaging techniques. Initially, studies focused on the asymmetry of the surface anatomy of the temporal regions (for a recent review, see [24]) and the corpus callosum [25,26]. As a whole, these early morphological studies pointed to an atypical pattern of reduced asymmetry in the usually most asymmetrical cortical regions, such as the planum temporale, a region just posterior to Heschl’s gyrus, and known to host the associative auditory cortex [27]. However, numerous individual exceptions make it hazardous to infer rules from these observations [28]. More recently, advances in the MRI technique have yielded new information, which is remarkably consistent with the functional data. Generally speaking, it appears that it is the same areas described as under-activated in functional imaging that have been reported as presenting a regional decrease in cortical thickness [29].

Using the Diffusion Imaging method (DTI-MRI), to track white matter structuration anomalies, several groups independently found converging evidence of impaired organization of the long white matter fascicles uniting anterior and posterior parts of the brain, mainly in the left hemisphere. One of these is the *arcuate fasciculus*, a horseshoe-shaped white matter tract that links the auditory areas and more generally the posterior sensory cortex to the lower frontal regions, including Broca’s area (Figure 2). Several studies have shown that impaired fibre organisation in this tract is a strong predictor of dyslexia, even in pre-school children, and is correlated to these children’s scores on phonological tasks [30]. The latter observation provides another argument in favour of the precedence of these abnormalities to any influence of reading and their role in the subsequent occurrence of the disorder. One of these studies, following a cohort of children with and without a family history of dyslexia [31], has shown that children with a genetic risk of dyslexia have disrupted white matter microstructure in the arcuate fasciculus as well as a protracted developmental trajectory between ages 5 and 12, strongly suggesting a genetic factor. In addition, another white matter tract, the inferior occipito-frontal fascicle, deemed to be significantly involved in the orthographic aspects of reading, has also been repeatedly found to be miswired and would be further influenced by the paternal reading level [32].

It is noteworthy that nearly all studies using DTI to investigate a possible impairment of subcortical fibre tracts in dyslexia have converged on the Arcuate Fasciculus, whose importance as the main gateway between Broca’s and Wernicke’s areas seems to consecrate dyslexia as a language-based condition [34], which can ultimately be attributed to a disconnection between a phonological processor situated in Broca’s area and phoneme representations stored in the superior temporal cortex [35]. More precisely, phonological dyslexia can be viewed as a specific failure of the left inferior frontal region (Broca’s area) to use otherwise intact information from phonemes stored in the temporal cortex, therefore causing a sort of decoupling between phonemic perception and phonological production [36,37].

Besides these findings relating mostly to the left hemisphere, stronger connections in right-lateralized white matter tracts, such as the superior longitudinal fasciculus, have been found in those dyslexic children who showed greater improvements in reading [38] as well as in those children with a family history of dyslexia who went on to become non dyslexic [33], suggesting that these right-lateralised white matter pathways may play an alternative or compensatory role in reading in children with dyslexia.

Converging results were obtained by magnetoencephalography in a study [39] exploring the coherence of neural oscillations between different brain regions: in subjects with dyslexia, there were inadequate connections between the right auditory cortex and left Broca’s area. In the same way, a Swiss team [40] investigated connectivity between the VWFA and various other cortical areas, using fMRI during an orthographic task, and showed that, contrary to normal controls, dyslexics fail to activate Broca’s area in response to the activation of the VWFA, yielding an authentic functional disconnection.

Finally, a number of studies have started dealing with the topic of connectivity in dyslexia from a broader perspective, that of connectomics [41]. With the help of new functional imaging techniques such as resting-state functional connectivity and adequate statistical tools, it has been demonstrated that dyslexics differ from typical readers in the temporal trajectories of functional connections between the phonological processors in the left inferior frontal cortex and the sensory centres in the inferior and lateral temporal areas [42]. Finally, studies using this type of general approach reached the conclusion that the activation of a number of networks not directly involved in reading is also affected in dyslexia, in particular those concerned with cognitive control and attentional processes such as the fronto-parietal and dorsal (DAN) and ventral (VAN) attentional networks [43].

In brief, the last ten years have seen an impressively growing amount of work whose conclusions would all seem to suggest that a lack of connectivity between regions more or less directly involved in the mechanisms of reading would appear to be the main anatomical and functional signature of phonological dyslexia. As will be argued in the following paragraphs, a similar explanation based on impaired connectivity, although mainly documented for phonological dyslexics, might also hold true for other forms of dyslexia as well as some of its comorbid neurodevelopmental disorders.

### 3.2. Brain Correlates of Attentional and Visuo-Attentional Deficits in Dyslexia

The brain imaging findings in this type of dyslexia are radically different from those of the “linguistic” subtype: they involve bilaterally the parietal areas, in regions known to be activated in various attentional tasks such as a flanked-letter categorization task [44], assessing visual attention mechanisms involved in multi-letter processing (Figure 3 and Figure 4).

Typically, these dyslexics fail in a task where they have to orally report a succession of five letters briefly presented horizontally on the computer screen, making a greater number of errors than phonological dyslexics (and also more than typical readers). The same authors [45] (Figure 5) showed, when proposing this kind of task to their dyslexics and controls under the MRI procedure, that dyslexics also fail to activate both (mainly right), superior parietal lobules. Interestingly, both inferior temporal cortices, including the left-hemisphere VWFA, are also strongly activated in controls but not in dyslexics, which may be interpreted as a disconnection between superior parietal/attentional and inferior temporal/visual systems.

Although strongly disputed by proponents of an exclusively phonological origin of dyslexia [46,47], the existence of this type of visuo-attentional mechanism in dyslexia has won occasional support [48] along with several independent demonstrations that specific training of attentional competencies may reverse brain anomalies associated with dyslexia [49], perhaps by reinforcing white matter connectivity between posterior visual and frontal executive networks [50]. Moreover, at least two studies from Germany (i.e., reading and writing in a transparent language) have mentioned the existence of non-phonological subtypes of dyslexics in which dyslexia is described as being probably attributable to an attentional mechanism [51,52] and where brain changes pattern is different from standard phonological dyslexia. Overall, it is tempting to hypothesize that, in this type of dyslexics, the core functional problem lies in impaired connectivity between the left-hemisphere temporo-occipital cortex and the visuo-spatial/attention-specific network in both hemispheres. If so, it would follow that the two anatomo-functional forms of disruption may frequently coexist in the same brain, yielding a mixed pattern of clinical impairments, both phonological and attentional, a situation frequently encountered in clinical practice.

### 3.3. The Dyspraxic Form of Dyslexia: The Mysterious Contribution of Motor Brain Structures to Reading Acquisition and Its Impairment

The existence of motor signs and symptoms in children with learning disabilities is a well-known and widely accepted observation in paediatric practice. Various so-called “soft neurological signs”, although initially described in the context of the somewhat outdated concept of minimal brain damage, have been convincingly revisited by Nicolson and Fawcett [53] in their well-known “cerebellar theory of dyslexia”. Pointing out the analogy between these motor symptoms, which they found in up to 80% of dyslexic children, and symptoms classically reported in cases of cerebellar pathology, they proposed a theory according to which dyslexia would appear to result from a presumed dysfunction of the cerebellum. In both cases, indeed, disorders of time estimation, motor coordination, muscle tone, balance and a deficit in automation have been reported [54,55]. Beyond this analogy, the theory attempts to describe the mechanisms by which cerebellar involvement could lead to difficulties in learning to read, pointing to the key role of a deficit in all types of procedural learning, including the grapheme–phoneme conversion procedure.

One interesting aspect of this theory is that it is also believed to explain concurrent motor signs often reported in dyslexics, especially awkward handwriting and the frequent co-occurrence of dysgraphia. Moreover, the same authors also suggest that this deficit could lead to difficulties in phono-articulatory realisation [56], which prevents the development of sufficiently precise phonological representations, bridging the gap with phonological theories of dyslexia [57], and opening new avenues for therapy [58].

Likewise, Pernet et al. [59] concluded from a meta-analysis of earlier literature that “the cerebellum is the best biomarker of developmental dyslexia”. An early study with PET by Nicolson et al. [60] had shown a deficiency in fine motor control and weaker cerebellar activations in adult dyslexics than in control subjects in motor learning tasks involving movements of the left-hand fingers. Yet more recent studies of the cerebellum in dyslexics have generally failed to disclose specific differences from normal reading individuals, either in morphology [61] or in function [62].

By contrast, studies using more sophisticated functional connectivity approaches have recently provided positive cerebellar findings: one of them [63] demonstrated abnormally higher engagement of the bilateral cerebellum in dyslexics during an orthographic task, which was negatively correlated with literacy measurements. According to the authors, the cerebellum would seem to play a compensatory role in reading in children with dyslexia. Another recent study found decreased functional specialization within the cerebellum during reading tasks, with differences according to reading proficiency in cerebellar regions associated with motor, but not language processing [64]. Finally, a connectivity study using probabilistic tractography [65] disclosed specific differences between 29 reading-impaired children and 27 typical readers, where impaired readers were found to have greater fractional anisotropy (FA) in tracts connecting the cerebellum with temporo-parietal and inferior–frontal regions compared to typical readers.

Taken together, these results suggest that the modulatory effect of the cerebellum on various brain regions involved in reading may be impaired in dyslexics, as a part of their overall dysconnectivity pattern.

Accordingly, some connectivity impairments have been reported in children with both dyslexia and dysgraphia, probably the closest condition to the aforementioned dyspraxic type of dyslexia. The few studies of this type indicated a defect of connectivity between the cerebellum and motor cortex [66], or between the left VWFA and motor regions [67] an interpretation which is consistent with MRI findings in normally-developing children and in adults, showing that orthographic and motor processes occur in parallel and interact during the writing process [68]. One interesting hypothesis would be that just as phonological dyslexia may be viewed as a functional disconnection between Broca’s area phonological processors and temporo-occipital auditory and visual associative areas, dysgraphia could be conceptualized as a disconnection of motor areas from linguistic processors in the left hemisphere. Most dyslexics with this profile would also suffer from impairment of connectivity between the cerebellum and the reading networks, as well as between other motor areas and language areas. A similar view has been proposed, following an extensive review of the literature on reading and the cerebellum, by Alvarez and Fiez [69], who conclude that the cerebellum plays an indirect role in reading due to its connections with inferior frontal and inferior parietal regions, which are presumed to intervene in phonological processing, and which both converge on the inferior temporal region to influence orthographical processes (Figure 6).

## 4. Comorbidities and Associated Features

As noted above, one of the universally recognized features of dyslexia is that it almost never arises alone. Thus, children with reading difficulties will very often receive another additional diagnosis, whether relating to oral language, coordination, attention, or calculation disorders. Studies suggest that as many as 50% of individuals diagnosed with a neurodevelopmental problem suffer from more than one disorder [70]. Comorbidity or a co-occurring disorder seriously impacts outcomes and generates significant constraints on family and school life. Furthermore, it complicates diagnostic procedures and increases the burden in terms of family expenses as well as for the healthcare system.

### 4.1. Speech/Language Disorders

Oral language disorders, including so-called specific language impairment (S.L.I.) or developmental language disorder (D.L.D., [71]), have complex relationships with dyslexia [72,73]. Indeed, dyslexia is widely believed to originate in an anomaly of a component of oral language, namely phonology, even though it is generally recognised that the phonological disorder is not an absolute condition for a diagnosis of dyslexia. Likewise, a number of children who have had difficulties with oral language are considered to be at risk for dyslexia, but some children with DLD, even in severe cases, will not become dyslexic [74]. Nevertheless, it is believed that even minor defects in the development of oral language, whether it is a disorder of the articulation of phonemes (speech sound disorder or phonological production disorder) or a disorder of the lexicon or syntax, even though they do not significantly impede the child’s intelligibility, are risk factors for subsequent dyslexia [72].

In comparison with dyslexia, there are far fewer studies of the brain substrate of D.L.D. Unlike dyslexia where reports have mainly highlighted changes in cortical regions, several meta-analyses of D.L.D. (e.g., [75]) indicate volume changes in subcortical nuclei, i.e., the lenticular and caudate nuclei. However, these reviews have pointed out the inconsistency of results across different studies, some reporting a reduction [76] others an increase [77] in the volume or functional activation of various areas or nuclei. Likewise, abnormal lateral asymmetries of structure and/or function have been inconsistently reported [78]. Some have tried to contrast language and speech disorders, the former being characterized by an increased volume of grey matter in the right hemisphere and the latter in the left hemisphere [79].

Finally, just as with dyslexia, connectivity studies have yielded the most relevant/promising results. For instance, at least two studies using DTI tractography have found impaired white matter microstructure in the form of decreased anisotropy in both dorsal and ventral language systems, i.e., respectively superior longitudinal/arcuate fasciculus, whose role is known to be the mapping of auditory speech sounds to articulatory (motor) representations and also processing complex syntactic structures [80], and inferior fronto-occipital (IFOF) and inferior longitudinal fascicles, involved in mapping sound-based representations of speech to conceptual representations ([81,82]. One unexpected yet consistent finding across these studies is that of a rightward asymmetry of the IFOF volume in DLD compared to controls (Figure 7), which can be interpreted either as a compensatory reliance on right-hemisphere processes or a result of impaired connectivity in the left ventral system, or both. In the dorsal network, in contrast, reduction in tract size seems to be specific to DLD since it is not found in language disorders associated with autism [83]. In addition, a study reported a lower volume of the right inferior longitudinal fasciculus, another part of the ventral language network, specifically in individuals with coexisting language and reading impairment [84].

### 4.2. Calculation Disorders

Another important comorbid association is the link between dyslexia and dyscalculia, which invariably will create a challenge in terms of academic progress, as it is widely acknowledged that good mathematical skills guarantee better integration and acceptance of the disorder [85].

Landerl et al. [86] studied four groups of 8–9 years olds: control subjects, who performed well in reading and numeracy, subjects only presenting dyscalculia, subjects only presenting dyslexia and children with a combination of the two. Overall, dyscalculics and those with both disorders behaved similarly and significantly differently from dyslexics and controls, with a tendency to treat small amounts in a serial and non-simultaneous manner, which some (e.g., [87]) consider the underlying disorder causing dyscalculia (impaired subitizing). Two more recent studies [88,89] of dyslexia/dyscalculia comorbidity reached similar conclusions, namely on the one hand the existence of cognitive risk factors common to both conditions (working memory, attention, executive functions) and secondly a signature specific to each of them, the phonological disorder in dyslexia and number sense deficit in dyscalculia.

Just as for other types of learning disorders, the use of diffusion MRI has provided valuable clues to understanding the brain substrate of dyscalculia, here again mainly in terms of disrupted connectivity between the parietal lobe, especially the region of the intra-parietal sulcus, the well-documented site of magnitude representation, and various sensory and language cortical zones.

A study [90] involving twenty-three dyscalculic children aged 7 to 9 years, suggested a central anomaly in the deep white matter of the right temporoparietal region, a region which is located on the path of two important tracts, the inferior longitudinal and fronto-occipital fascicles. The two bundles link posterior regions, such as the inferior temporal cortex, with anterior zones, in particular the inferior frontal areas. In addition, the posterior fibres of the callosum (connecting the two temporo-occipital regions) also cross the midline near this zone. This notion of disconnection in dyscalculia has also been mentioned more recently in a study [91] which demonstrated a decrease in anisotropy, reflecting degraded connectivity, in various sectors of the superior longitudinal fascicle bilaterally, more precisely on the left near the parietal cortex and the pre-central cortex, and on the right near the Insula. The significance of these anomalies is not discussed, but the authors emphasize the fact that the upper longitudinal fascicle conveys information relevant to numerous cognitive functions and that its deterioration would seem to suggest a more general cognitive involvement in calculation disorders.

More recently, a survey [92] of the available literature about brain and arithmetic across different developmental pathologies, also came to the conclusion that calculation disorders may be linked to damaged white matter tracts connecting distinct parts of the arithmetic network. Authors discuss the involvement of the left anterior part of the arcuate fasciculus as a common anatomical substrate of dyslexia and dyscalculia, its morphology being positively correlated with addition and multiplication [93] but not with subtraction and division, which suggests this bundle plays a role in fact retrieval as well as in reading. Interestingly, a study of training-induced changes in arithmetic in relation to white matter morphology [94] showed that the part of the SLF that connects the frontal and temporal regions predicted the learning gains in addition and subtraction. 

Finally, special mention must be made of developmental Gerstmann syndrome (the co-occurrence of dyscalculia, dysgraphia and somatognosic disorders, including digital agnosia [95]), whose existence has been disputed, but which not only draws attention to the link between calculation and digital gnosis, but also draws the parallel between a classic neurological syndrome in adult lesion pathology and a developmental syndrome in children. Interestingly, it has been suggested that Gerstmann syndrome could result from a disconnection between various areas which are themselves otherwise intact [96].

### 4.3. Coordination Disorders

An extensive review of the relevant literature [97] found 11 studies exploring coexisting dyslexia and motor impairments and/or developmental coordination disorders (DCD) (dyspraxia), among which 36% had both diagnoses. Conversely, out of seven studies of children with DCD, 56% had significant reading difficulties and/or a diagnosis of dyslexia.

This specific case of dyslexia/DCD comorbidity has been the subject of a debate surrounding the aforementioned cerebellar theory of dyslexia championed by Nicolson and Fawcett [98]. These authors found 81% to 84% of dyslexic children to perform poorly in a balance task, either in dual-task or in closed-eyes conditions, compared to 13% to 16% in control children, which they interpreted as reflecting subtle cerebellar dysfunction. With a comparable procedure, Fawcett et al. [99] also showed that dyslexic children aged 10, 14 and 18 remained stable for a shorter time when they stood with their eyes closed and feet together, and that they had difficulty recovering their balance after a slight push in the back. These balance problems were not found in all studies, however, especially when controlling for the presence of attention problems [100,101,102,103]. For these authors, balance problems were a sign of the association of dyslexia with attention deficit disorder with or without hyperactivity (ADHD) or developmental coordination disorder (DCD), a position also held after a systematic study of 58 dyslexic children by Chaix et al. [104] who found rates as high as 57% of more or less severe motor symptoms, which they ascribed to coexistent attentional deficits as assessed on sustained and selective attention tasks.

As already mentioned, there is far lesser imaging data available for DCD than for dyslexia. Moreover, it is subject to the almost inevitable confounding bias caused by the association with ADHD. Therefore, almost all imaging data has been obtained in cohorts of patients receiving both diagnoses together. In this context, the pathological findings relevant to DCD mainly concern brain structures involved in motor or sensory–motor processes, such as the corona radiata and cortico-spinal tract, cerebellum and motor cortex [105,106]. A recent diffusion-weighted imaging study [107] confirmed this general view, showing altered white matter microstructure in several tracts involved in motor processes, including cerebellar peduncles, which would seem to be one of the most robust findings in this context. In addition, the pattern of diffusion parameters in children with DCD (axial but not radial diffusity) suggests that axonal development may be disrupted. Finally, a resting-state functional connectivity study provided arguments in favour of a functional disconnection between sensory–motor and posterior cingulate cortices, thought to play a role in the inefficient motor learning seen in DCD [108].

### 4.4. Attention Disorders with and without Hyperactivity (ADHD)

ADHD is four times more common in children and adolescents with reading and spelling disorder, and its prevalence in children whose reading and spelling disorder has already been diagnosed is 8–18%. The coexistence of both conditions is known to greatly increase the burden of each condition considered in isolation [109]. For example, it has been shown that children who face the cumulative problems of both disorders are at greater risk of academic failure, psychosocial consequences, and poor long-term outcomes that persist into adulthood [110].

ADHD itself has been related to impaired connectivity within specific brain networks known to depend either on the dopaminergic fronto-striatal system, or on the dorsal and ventral attentional networks (DAN/VAN), even more so in the case of comorbidity with fine motor control deficits [111]. Several DTI and functional connectivity studies [112,113], have shown that various white matter paths are abnormally organised in ADHD children and adolescents, especially the superior longitudinal fasciculus, cortico-spinal tracts and cortico-striatal connections, the latter being regionally linked to clinical symptoms (such as orbito-striatal anisotropy found proportional to inattention symptoms, while the uncinate fasciculus is linked to impulsivity [114,115]).

More recent methods of global connectivity assessment have shown the particular role in ADHD of connections with and within the default mode network (DMN), whose activity, which alternates with the cognitive control network in opposing directions according to attentional demands, is presumed to be disrupted in ADHD [116].

## 5. Toward a Unified Cross-Modal Impairment Explanation of Dyslexia and Related Disorders

As a whole, this overview of the recent relevant literature about the brain substrate of specific developmental disorders provides an overall vision of the brain mechanisms at work as an ensemble of *developmental disconnection syndromes*, whereby various brain systems that must co-activate during a specific period in order for the function to be acquired normally, will fail to do so because of atypical development of specific white matter bundles. The idea of dyslexia and related disorders as disconnection syndromes is not new. Already, Norman Geschwind, a pioneer in disconnection models in neurology, intuited that dyslexia could be explained by a visual–auditory disconnection: “*In an illiterate society, a lack of visual*–*auditory associations would not seriously inconvenience anyone except in unusual situations; literacy makes this ability highly important. Other cross-modal association deficits may exist but might never be detected because they cause so little disturbance*” [117].

This has been clearly illustrated for dyslexia in the acquisition of grapheme-to-phoneme conversion processes when learning to read, but could also be applied to other conditions such as dysgraphia and dyscalculia as well.

Several separate groups have developed the idea that dyslexia could be more generally related to the inability of the brain to allow reciprocal mapping of stimuli of different natures, such as the visual image of a letter (grapheme) and its corresponding sound (phoneme). A Dutch group [118,119] used functional MRI in various perception conditions: auditory alone (sound), visual only (letter), and finally combined letter/sound conditions, either congruous (the sound and the letter correspond), or incongruous. The results show that the associative auditory cortex is specifically involved in these tasks and its activation is defective in dyslexics. Most importantly, controls have less activation for incongruous pairs, but not dyslexics, reflecting the inability of their associative cortex to process the incongruence of letter/sound correspondence. These findings recall the MacGurk effect, a phenomenon well known to phoneticians, where erroneous visual feedback (e.g., video of a face pronouncing “ga” when listening to the syllable /ba/), induces an auditory illusion where the subjects hear a third syllable (da), corresponding to the fusion of the two consonants into a third one. Dyslexic individuals, in contrast, do not experience the fusion [120], and this has been related to a failure to activate cross-modal cortical areas, both in the OT visual and the temporal auditory regions [121].

In a recent review of the relevant literature, Richlan [122] collected evidence from various sources and experimental settings suggesting that, in developmental dyslexia, a specific neurocognitive deficit in the crossmodal integration of letters and speech sounds can be shown to hinder the binding of orthographic and phonological information. Therefore, this will impede the emergence of a functional neuroanatomical brain system in the left ventral OT referred to as the “reading skill zone”, as well as other bilateral associative cortical regions in the temporo-parietal and frontal lobes, required for “fast, fluent, and seemingly effortless reading”. More generally, these observations suggest that the dyslexic brain has particular difficulty integrating orthographical and phonological representations, which could be the cause of the notorious reduction in fluency observed in dyslexia, especially for transparent orthographical systems such as German.

Finally, there is substantial evidence that the audio–visual integration impairment found in dyslexics is related to impaired anatomical connectivity in white matter tracts. One such study [123] found a direct correlation between audio–visual activity in the posterior STS and fractional anisotropy in the posterior region of the left arcuate fasciculus (AF) measured by diffusion tensor imaging, both being proportional to reading efficiency. This suggests that audio–visual integration relies on the integrity of the posterior part of the AF, probably through fibres connecting auditory and visual association cortices.

Altogether, these results complete the picture of a dyslexic brain mainly characterised by multiple atypical patterns of connectivity, each of them possibly responsible for each individual combination of symptoms reflecting the different forms of reading impairment and their associated comorbidities in the main above-mentioned cognitive domains. Within this theoretical framework, it will become clearer how such apparently different symptoms may share a common pathophysiology, and how each of them will benefit from interventions aiming to improve or enhance cross-modal connectivity, as will be described at the end of this article.


**Part two. It is only a matter of time: a comprehensive temporal perspective in dyslexia and related disorders**


The first part of this review article, which was devoted to general clinical considerations and brain imaging data, reached the conclusion that although possibly originating from different basic mechanisms and cascading effects of impaired connectivity in different brain regions, these clinical descriptions of specific learning disabilities, may have in common the notion of impaired integration of multimodal information processed in different parts of the brain.

This second part will explore the possibility that one crucial factor leading to this general impairment of the process of learning lies in the timing properties of the information that is processed. Indeed, without going into complex notions such as axon velocity and temporal aspects of synaptic transmission, which clearly fall outside the scope of the present review, the link between microstructural axon alterations, as reflected in DTI and tractography, and timing properties of brain tissue, including processing speed, has been amply demonstrated in animal as well as various human pathologies [124,125,126]. It is not surprising, then, that different forms of timing impairments are found in dyslexia.

The following paragraphs describe these different aspects of time processing in dyslexia and some of its related/comorbid disorders, both from clinical and experimental points of view, starting with the well-known “temporal processing deficit theory” of dyslexia and the clinical description of “dyschronia”, a syndrome of impaired time cognition often reported in dyslexic patients.

An exhaustive search for studies between December 2000 and December 2020 was performed through the Pubmed database. The keywords used in this search were:

(Dyslexia [title] AND (temporal processing [title/abstract] OR (time processing [title/abstract] OR rhythm [title/abstract]). A Preferred Reporting Items for Systematic Reviews and Meta-Analyses (PRISMA [127]) diagram summarizing the number of studies meeting the search criteria is shown in Figure 8.

## 6. Impaired Timing Processes as an Explanation for Dyslexia

### 6.1. The Temporal Processing Theory of Dyslexia

Temporal processing is a very broad concept referring to the way the brain manages two or more stimuli presented non-simultaneously. Initially proposed by its authors to account for language disorders in general, as an inability of the brain of these children to process information that is formed of brief elements received in rapid succession [127], it was subsequently extended to explain dyslexia and its related phonological deficit [128]. More precisely, these children would be unable to effectively and reliably process rapidly changing and serially ordered brief speech signals such as formant transitions, spectral noise associated with plosives, and differences in voice onset time (VOT) in voiced and unvoiced consonants. As already mentioned twenty years ago [1] and largely developed since then, this “temporal processing deficit theory” of dyslexia was based on the demonstration of deficits in dyslexics with temporal judgment tasks in different perceptual modalities: auditory (e.g., judgment of order of two heard tones or syllables), but also visual, and even tactile ones [128,129]. Furthermore, in the auditory modality, studies have shown that individuals with language and/or reading impairment also have non-verbal temporal processing deficits [130,131,132]. Others have challenged the hypothesis, arguing that reading disability results from the linguistic system failing to discriminate phonological representations if they are perceptually too similar, rather than from impaired temporal processing of sensory signals [133,134,135]. In addition, some studies have shown that temporal processing deficits are not present in all individuals with dyslexia, and that impairments in one modality do not necessarily co-occur with those in another modality [136,137]. Finally, critics have mainly focused on the spectacular results obtained using a method of rehabilitation called Fastforword^®^ [138,139], which primarily targets rapid auditory processing and which produced mixed results during evaluation by separate groups [140]. Overall, the effectiveness of FastForword^®^ was confirmed but did not differ from that of other more traditional phonic exercises. This does not undermine the legitimacy of the temporal hypothesis, which has been further supported by studies with functional MRI [141], showing that dyslexic children who underactivate the left hemisphere language network during phonological tasks, recover a normal pattern of brain activity in their language areas after a few weeks of training with Fastforword^®^. Accordingly, a study comparing left prefrontal activation for rapid transitions relative to slow ones in dyslexics showed an improvement for rapid transitions after similar training [142].

In the same way, our group has carried out a series of studies in dyslexic children using the mismatch negativity ERP method to contrast various features of the content of speech [143]. Children were presented with a sequence of syllables that included standards (the syllable “Ba”) and deviants in vowel frequency, vowel duration and Voice Onset Time (VOT) that were either close to or far from the standard (Small and Large deviants). The electrophysiological response allowed us to explore pre-attentive processing of these various phonetic traits. Dyslexics did not differ from controls for the frequency deviants, demonstrating normal pre-attentive processing for this variable, which did not involve temporal processing. In contrast, they showed an abnormal response on the other two variables, both of which are dependent on the temporal characteristics of the stimuli. In a subsequent study [144], dyslexic children received the same ERP protocol before and after six months of music/rhythmic or painting training (twice a week for 45 min for a total of 300 h of training). Similarly, only the two time-dependent variables were improved after music (but not painting) training, further supporting the argument for the temporal nature of their linguistic deficit.

One recent and important elaboration on Tallal’s temporal processing theory of dyslexia has arisen from a series of studies focusing on multisensory temporal paradigms, beyond the demonstration of temporal deficits in one given modality (mainly auditory, but also visual and tactile), i.e., using temporal order judgement or temporal processing acuity in audiotactile, visuotactile or audiovisual stimuli combinations [145]. Crossmodal temporal order tasks have been shown to activate a vast parieto-temporal network, suggesting an interplay between spatial and executive functions [146]. In temporal acuity experiments, subjects had to judge the simultaneity of a visual or auditory stimulus co-occurring with variable temporal precision with a tactile indentation perceived under the index finger. In almost all the combinations, young dyslexic performed less well than controls. Moreover, their performance was related to phonological awareness. These results strongly support an important contribution made by a supra-modal temporal processing deficit in dyslexia. Similar results have been obtained in children and adults with autism spectrum disorders, prompting some to consider similarities between the mechanisms of autism and dyslexia [147]. Finally, the temporal processing deficit theory, although partly fallen into disuse, still deserves credit for having encouraged the scientific community to consider timing deficits as a possible etiologic factor in dyslexia and related disorders [148]. Moreover, compared to previous purely phonological theories of dyslexia, it had the advantage of providing potential explanations for some features associated with the reading deficit, including often reported syntactic as well as problems in the memory of order and possibly more general time processing difficulties often encountered in a number of patients with dyslexia or other learning disorders, under the concept of “dyschronia”.

### 6.2. Dyschronia: A Frequent Yet Overlooked Syndrome of Time Processing Impairment

One fairly common feature of cognitive impairments in children with specific learning disorders is a significant and long-lasting tendency to struggle with temporal notions and representations, such as situating themselves in time, in all its dimensions (hours, days, weeks, etc….), and generating an accurate representation of durations and time intervals, a condition sometimes referred to as dyschronia [149]. For example, it is frequently noted (most often by their own parents) that children experience a vague understanding of time passing, or demonstrate serious misinterpretation of an event’s duration, which results in significant limitations in numerous daily activities which require a good perception/representation of time. Thus, temporal notions, such as time-related vocabulary terms (weeks, months, seasons; yesterday, today, tomorrow…), or the accurate estimation of a time interval (saying how long a movie is, or how long it takes to get dressed in the morning) will be more or less clearly mistaken, as if the child lacked a stable duration referential to rely on. During school activities, this will often have vexing consequences in addition to those specific to the reading or writing difficulties, such as confusion between syntactic forms, especially tenses, or misunderstanding the chronology between, say, Prehistory, Antiquity and the Modern era.

Overall, it is tempting to try and unify these apparently heterogeneous manifestations of temporal cognition disorders under one unique concept of “inner time sense” [150], reflecting the more or less conscious (indeed probably largely implicit) awareness of time passing, or temporally locate an event in the recent past, as well as correctly ordering successive stimuli or events. Although in our experience this feature is more frequent in children of the third “dyspraxic” profile, it may also be present in the other two, thus compromising the acquisition of several skills as well as autonomy, sometimes severely. Thereby, dyschronia participates significantly in the cognitive impact on the quality of life and academic achievement of dyslexic children.

### 6.3. Time Perception, Time Processing and the Brain

A study by Casini et al. [151] has specifically explored the hypothesis that dyslexia is caused by a domain-general temporal processing deficit, as opposed to an auditory-specific deficit. They devised several auditory and visual, verbal and non-verbal tasks where participants had to discriminate or estimate the duration of various stimuli. Compared to controls, dyslexic children made significantly more estimation errors in all modalities, a result the authors interpret as suggesting a deficient “internal brain clock”, with reference to the classical “pacemaker-accumulator model” [152]. According to this model, a pacemaker produces a series of “pulses”, which are analogous to the ticks of a clock, and the number of pulses recorded during an interval represents experienced duration. The greater the number of pulses, the longer the subjective estimation of duration will be. Interestingly, there were strong correlations between these temporal processing abilities and dyslexics’ reading and phonological abilities, but no mention was made of their day-to-day awareness of time nor of the presence or absence of a clinical syndrome of dyschronia. Therefore, it would be of particular relevance to discuss the possible candidates as a brain locus of the “internal clock”. Despite being suspected for a long time [153], a cerebellar origin of dyschronia in learning impaired children has never been formally established, even though historically the cerebellum was the first structure whose timing properties have been demonstrated [154]. Indeed, there are numerous arguments for attributing the function of the brain’s internal clock to the particular organisation of the cerebellum, especially to the olivo-cerebellar system [155]. Nowadays, although less often cited for its timing properties [156], the cerebellum remains the most likely candidate for ensuring the timing of motor activity and more generally of sensory–motor integration [157]. Overall, there seems to be a consensus on the role of the cerebellum as a “timing machine” probably through its interactions with the basal ganglia [158]. The respective role of the cerebellum and basal ganglia in motor timing is still uncertain. For some, the basal ganglia perform basic timing processing and the cerebellum, through its reciprocal connections with the basal ganglia, performs subsequent timing adjustments or other complex timing processes [159].

## 7. Dyslexia and other Learning Disorders as Specific Time-Dependent Manifestations of Altered Connectivity

### 7.1. Rhythm, Dyslexia, and the Brain

The notion that dyslexic children may often experience more or less serious difficulties in various tasks requiring them to reproduce a rhythm or follow the pace of a metronome is a rather old one. In the mid-20th century, the French physician Mira Stambak [160] already demonstrated the obvious difficulties encountered by some dyslexic children in a task where they had to reproduce a series of white noises produced by the examiner and unseen by the patient, for instance by tapping with a pencil. Later on, Wolff [161] reported excessive anticipation by dyslexic students of the signal of an isochronic pacing metronome, in intervals that were two or three times as long as those of age-matched normal readers or normal adults. In addition, they had difficulty reproducing simple motor rhythms by finger-tapping, and also reproducing the appropriate speech rhythm of linguistically neutral nonsense syllables. In fact, various aspects of temporal cognition are affected in dyslexics, from the perception of duration [153] to difficulty in the production of the order of phonemes or letters [131] to simply producing a movement that is synchronous with a beat arbitrarily given by a metronome [162]. Indeed, the difficulties experienced by dyslexics in this type of task are to a certain extent proportional to their performance in reading [163].

The same was also found in a group of children with specific language impairments, here again proportionally to language and literacy outcomes [164]. However, the contribution of rhythm seems to differ between specific language and reading deficits. For example, in a study where children from each group received either a humming task (syntactic comprehension of a sentence was facilitated by associating its prosodic contour), or a non-word/musical stress association (where trisyllabic non-words had to be associated with three-note musical sequences with different musical stress), the language-impaired children only failed the first task while the reading-impaired were less efficient on the second one [165].

Recently, several studies have confirmed the crucial role of rhythm in reading by showing a strong correlation between the ability to perceive musical and speech rhythm and reading ability [166,167] as part of a new topic in psychology, often called “rhythm cognition” [168].

Finally, studies (e.g., [169]) suggest a relationship between the rhythmic perception abilities of 6-year-olds and their performance in syntactic tasks, suggesting a strong link between the oscillatory activity (see below) of the cortical areas of language and the linguistic function itself. One important contribution in this regard has been made by a group in Lyon [170,171] who showed through an ingenious protocol that syntactic processing may be facilitated in dyslexics by exposing participants to either regular or irregular rhythms for a few seconds just prior to the linguistic task. The results quite amazingly show that this short exposure alone is sufficient to significantly enhance the subsequent linguistic performance (judgment of syntactic correctness of a sentence). Conversely, patients performed less well when previously primed with an irregular rhythm. In other words, passive priming with regular or irregular rhythms has a preparatory effect on language processing mechanisms, most probably via neural entrainment of specialised brain circuits by the regular rhythm but not by the irregular one. Finally, a study of adult dyslexics has found that this neural entrainment is less efficient in dyslexics compared to controls, with only the latter being able to extract regularities from irregular rhythms [172].

### 7.2. The Temporal Sampling Deficit Theory of Dyslexia

As initially proposed by Poeppel [173], the brain (probably the lateral cortical zones of the two hemispheres) is capable of processing the various traits of human speech by matching them with the spontaneous oscillations that characterise cortical physiological activity. These oscillations have long been identified according to their specific frequency, probably substantiated by different neural networks whose frequency will correlate with the various temporal constituents of speech through a coupling mechanism, i.e., delta (≈1.5 Hz) for prosody, theta (≈7 Hz) for syllabic rhythm, gamma (≈40 Hz) for the phonological structure. Through this sampling process, the sound stream becomes segmented into discrete chunks that constitute the basic coding elements for subsequent neuronal computation, in particular, for accessing the semantic and syntactic levels [174]. Moreover, these processes appear to be sequential and hierarchical, with the slowest activity nesting the faster ones.

In dyslexia, this process appears to be specifically impaired, due to poor sampling of the speech signal and consequently defective synchronisation with speech sounds. According to *the temporal sampling theory of dyslexia* proposed by Goswami [175], deficits in temporal sampling and inefficient phase locking at one or more temporal rates could explain abnormal phonological development in children with dyslexia across languages.

Goswami further argues that individual differences in phonological processing in language should be related to individual differences in non-linguistic musical tasks based on patterns of beat distribution, namely the succession of strong and weak syllables, or the « metrical » structure of speech and prosody. Accordingly, children with developmental dyslexia have auditory perceptual impairments in sound rise time perception that are believed to affect their sensitivity to metrical structure [176]. One preliminary conclusion is that the initial problem would be an inefficient phase-synchronization of slow frequencies in the delta and theta ranges to prosody and syllable rates respectively, possibly originating in the superficial cortical layers of the right auditory cortex. As slow frequencies are thought to regulate faster ones, an abnormal phase-synchronization to speech input at low frequencies may result in abnormal oscillations at higher frequencies, thus altering the proper encoding of speech at the phonemic level. This altered connectivity would prevent stimulus-driven spike trains from layer IV, from being properly transformed by gamma and theta oscillations.

Specifically, this impaired entrainment of delta range oscillations in the right auditory cortex then appears to affect the oscillatory activity of the left IFG, negatively impacting the correct identification and manipulation of the phonological categories stored in left posterior temporal regions [39,177].

Although attractive, this theory suffers from the same limitation as all explanations focusing on phonology to explain dyslexia, namely its inability to account for not only the existence of clinical subtypes but also the numerous comorbidities of dyslexia. Moreover, a recent survey of the literature [178], advocating the so-called “oscillopathic” origin of dyslexia, reports a variety of anomalies at different frequencies and different localisations, rendering a synthetic view rather problematic, at least according to current knowledge.

In the visual modality, it has been proposed that the same principle of hierarchical nesting of networks with different oscillation frequencies could serve to focus attention on individual graphemes [179] and that dyslexia could result from a primary defect in visual rather than auditory cortical oscillations. Moreover, the same mechanism would also serve to shift attention, along with associated eye movements, from one fixation to another, to ensure smooth and accurate gaze tracking during reading. As hypothesized by Vidyasagar [180], in the perspective of a more general timing deficit in dyslexia, encompassing both early visual and auditory pathways, “the most significant timing aberration may well be that in the visual system, since reading begins in the visual system and precedes the phonological process”. Arguably, the difference in phonological abilities between typical and impaired readers may also be a consequence of the “immense difference in reading experience” such that “children with developmental dyslexia read in a year the number of words that typical readers of the same age read in a couple of days”.

Finally, other neurodevelopmental disorders have also been considered, albeit more rarely, from the point of view of asynchrony of oscillations between distant brain networks, especially ADHD and autism.

### 7.3. Abnormal Oscillatory Function in ADHD and Autism

In ADHD, a large body of literature has recently been devoted to the study of the oscillatory behaviour of neural networks in the so-called DMN (default mode network), known to play complex and multiple roles at the interface between cognition and consciousness. For example, they are specifically activated during mental states of mind-wandering (daydreaming), when attention is relaxed but with the individual still in a state of intact vigilance [181]. In opposition of phase with its executive counterpart in the dorso-lateral prefrontal cortex (PFC), the very low oscillations in the DMN (below 0.1 Hz), present in the resting state, will cease when the individual is engaged in an active process of thinking or any intention to initiate an action. Inappropriate regulation of this network has been associated with inattention, a core characteristic of attention-deficit/hyperactivity disorder (ADHD), especially in the form of abnormal intrusions of DMN activity during attention-demanding contexts, as well as insufficient suppression of the dorsal PFC. In both cases, at a neurofunctional level, the anomaly appears to take the form of aberrant timing of oscillatory activity causing abnormal connectivity between the two systems.

Likewise, it is conceivable that in cases of co-occurring dyslexia and ADHD, the inefficient phase-synchronization of slow frequencies characteristic of the dyslexic brain would even be augmented by the superposition of inefficient control from the DMN/PFC couple.

Repeated demonstrations of timing deficits in ADHD adults and children [182,183], who generally over-estimate durations and intervals whatever the modality used (auditory or visual), are generally interpreted in the framework of the “clock-accumulator model” as impaired transmission of internal clock information to the locus of a presumed accumulator. Here again, demonstrations of “laboratory results” of impaired timing function have never been correlated with patients’ day-to-day subjective utilisation of time markers, or to the presence of a comorbid coordination disorder or dysexecutive syndrome. Nevertheless, auditory timing discrimination deficits have been demonstrated in children with developmental coordination disorder (DCD), which the authors see as grounds for using auditory–motor exercises to treat DCD [184].

Finally, although arguably outside the scope of this review, the issue of temporal processing impairment in autism and autistic spectrum disorder (ASD) is worth mentioning here due to multiple demonstrations of both temporal cognition impairment and impaired brain connectivity and oscillatory coherence in the social processing network [185,186], especially decreased long-range synchrony and increased posterior local synchrony, with each effect limited to a specific frequency band and related to impairments in social functioning. Remarkably, just like other conditions that are reviewed in the present article, autism has repeatedly been considered a time processing disorder (for a review, see: [187]), with impairment in different dimensions of time processing such as: time perception (experiencing and assessing the length and passage of time, including interval timing), time orientation (awareness of the day, date, month and year, as well as the ability to situate events on a timeline), and time management (the ability to prioritize one task among others, to order a set of tasks, and to allocate the optimum amount of time to each task [188]). It is noteworthy that, just as in the case of dyslexia, ASD has been associated with decreased susceptibility to the McGurk effect [189], or poorer judgement of simultaneity in an audiovisual task, suggesting a specific deficit in crossmodal sensory integration. This deficit would be potentially causal to social cognition impairment, due to a cascading effect from multisensory processing to speech perception [190]. This observation of the crucial role of multisensory timing in ASD has prompted researchers to propose to integrate a systematic slowing of information to interventions as an aid to improve social cognition in ASD children [191], not unlike that once proposed for dyslexic children in line with the temporal processing deficit theory.

## 8. Toward a General Temporal Theory of Specific Learning Disabilities: Reconciling Impaired Connectivity and Inaccurate Timing of Cortical Oscillatory Function

In an attempt to integrate the various facets of temporal impairment in dyslexia and related disorders, it would seem reasonable to question the similarity and possible commonality between facts and observations as varied as the clinical occurrence of dyschronia, experimental evidence of impaired perception and reproduction of elementary rhythms or even the changes observed in linguistic or non-linguistic features of dyslexia by manipulating any temporal attribute of the individual’s environment. Obviously, success in this endeavour will determine our ability to gain a new understanding of an area of neurology that remains mysterious, but also to find new ways to alleviate the considerable burden it represents worldwide for individuals as well as societies.

The first thing to note concerns the concept of learning itself. Looking at the most frequent areas of cognition that are involved in the different syndromes grouped under the umbrella term “learning disorders”, and perhaps the more general term “neurodevelopmental disorders”, it is tempting to reduce them to their smallest common denominator which could be stated as follows: (1) an area of knowledge where several different brain systems (at least two), which may be topographically distant from one another, are required to start the acquisition process; (2) an area where the intervention of these different cognitive requirements and/or neural systems must follow precise temporal rules to ensure successful acquisition; and (3) an area where repetition is crucial to the harmonious development of function. Reading, writing and calculating all fulfil these general requirements. Concerning condition (1), it is conceivable that since the ability to combine the individual codes of distinct modalities (such as visual/auditory; perceptual/phonological; spatial/linguistic) is crucial, successful learning would be totally contingent on the necessary integrity of the white matter tracts that ensure their reciprocal connections. It would also follow that the most vulnerable functions in this regard would be those requiring altogether temporal precision in the co-occurrence of the specific information (as expected in a general Hebbian learning mode), and repetitive exercise of the emerging function during a limited critical period (probably the first two elementary school years in most cases). A similar reasoning applied to other neurodevelopmental disorders is conceivable, although far more hypothetical in the current state of knowledge.

Taking for granted this general framework, as efficient connectivity is thought to depend on both timing precision in information processing and repetitive exposure to crossmodal associations, it is not surprising that impairment of the learning process will be found in association with both timing imprecision and atypical connectivity, with clinical dyschronia occurring as a macroscopic manifestation, or alternatively an indirect witness, of an underlying elementary and ubiquitous temporal processing weakness, which is itself related to a general, multisystem developmental disconnection mechanism.

In this respect, the oscillatory explanation of dyslexia is intended as a working model to be expanded to other areas of the cognition of learning, the integration between orthography and phonology being perhaps only the most immediately accessible illustration of a much broader phenomenon, with a wide range of applications. As already envisaged for the visual modality in reading, it is possible that the oscillatory properties of various neural systems throughout the brain connectome may determine the normal or atypical functioning of any pair of networks whose combined action is required to enter into any learning process. The example of social cognition, and its emerging learning rules, is one of the most fascinating as well as promising ones in terms of its potential applications.

Whether the primum movens in dyslexia is to be searched in the neurogenesis of white matter tracts or in the timing deficit itself, or even in the combination of both, cannot reasonably be discussed here, but it could conceivably be argued that the causality issue is less important than the possibility of developing therapeutic strategies whose potential efficacy would be a real step forward in an area of neurology where evidence-based therapy is limited and most likely still in its infancy.

## 9. Music Learning as a New and Potentially Helpful Therapeutic Tool for Improving Dyslexia

At least two independent studies have compared adult dyslexics both with and without an experience of musical practice and non-dyslexic adults and found that dyslexic musicians outperform dyslexic non-musicians on several reading, phonological and auditory memory tasks [192] and even non-dyslexics on some rhythmic tasks [193]. More generally, musical training has been associated with improvement in various cognitive skills as well as academic achievement, including reading and language (for reviews: [194,195]).

Apart from observational data about the general cognitive effect of musical experience in adult dyslexics, relatively few studies have prospectively explored this effect in dyslexic children. Overy [196] proposed musical activities to dyslexic children, which were designed to progress gradually from a very basic level to a more advanced level over a period of 15 weeks. The results showed a significant improvement, not in reading skills, but in two related areas: phonological processing and written transcription tasks. In addition, performance in transcription was significantly correlated with performance in a timing task. Bhide et al. [197] compared in poor readers the effects of a musical intervention to those of a rhyme training and phoneme–grapheme learning software of proven efficacy. In terms of phonological and reading improvements, there was no difference in the results between the two methods.

More recently, Flaugnacco et al. [198] compared the effect of systematic rhythmic training to visual art practice in a meticulous study of 83 Italian dyslexic children. During bi-weekly, one-hour training sessions, groups of 5–6 children were offered either musical training, focussing on rhythm and temporal processing or painting training, the latter following a program designed to promote visuo-spatial skills and manual dexterity as well as creativity. The results were very clear, with musical training having a greater effect on phonological and reading tasks, as well as on a rhythm reproduction task. The result of the rhythm production task has proven to be the best predictor of phonological awareness as measured by the phoneme fusion and phonemic segmentation tasks. In a methodologically similar study, i.e., comparing musical and non-musical artistic activities of comparable duration and intensity, Frey et al. [144] recently obtained ERP confirmation of the specific right and left hemisphere auditory cortex improvement in pre-attentive perception of a typically temporal feature of speech-sounds, the voice onset time (VOT), allowing differentiation between voiced and unvoiced consonants.

Among theories linking reading achievement and music training, one of the most popular ones emphasizes similarities between language (in general, and phonology in particular), and the quasi-linguistic properties of the musical code [199]. Accordingly, in dyslexic children, improvement following musical interventions has been found tantamount to that following phonology training [200] in accordance with the above-mentioned phonological-only account of dyslexia. Yet there is as much evidence that music and language are differently organised in the human brain, especially the vast literature about dissociated substrates for aphasia and amusia following brain damage (e.g., [201]). Another distinct interpretation of the effect of music training in dyslexics would be that it develops compensatory activation of the right-hemisphere temporo-parietal areas symmetrical to those under-activated in phonological dyslexia [202]. Other accounts of the effect of music in dyslexia have highlighted the attentional component of the musical or rhythmic exercises [203] or more generally an effect on executive functioning [204]. Likewise, there is a large amount of evidence showing that music training enhances working memory processes, and that this effect could be related to increased intra-hemispheric coherence in theta oscillations [205,206] leading to the conclusion that music training may modulate the cortical synchronisation of the long-range neural networks involved in verbal memory formation.

One strong assumption, then, to explain these observations comes from Diffusion Imaging data obtained in professional musicians, which showed that, in comparison with musically naïve adults, the former have larger and better fibre organisation in the arcuate fasciculus [207], precisely the same white matter tract that has mainly been found to be affected in dyslexic children and adults. Accordingly, several studies with Diffusion Tractography have shown that music training is able to significantly modify the size and anisotropy of the arcuate fasciculus, even over short training periods (a few days), especially if the rhythmic content of training is multi-modal, involving simultaneous visual and auditory input [208]. Assuming that the key feature of dyslexia, as well as perhaps of other associated conditions, is deficient connectivity between areas whose interaction is crucial to ensure basic learning, we can therefore reasonably presume that intensive training based on crossmodal integration, in addition to auditory and rhythmic aspects, (i.e., involving synchronous and repeated activation of auditory, visual and motor brain areas like playing on a keyboard, performing rhythmic body movements, or dance), will likely improve variables believed to reflect dysfunction in dyslexics.

Our group has recently developed a series of training tools and protocols, collectively termed “CMT” (cognitive musical therapy [209]), aimed at exercising and ultimately increasing the efficiency of such connectivity, with some encouraging results. After 6 weeks of intensive multimodal exercises systematically involving visual, auditory and motor components of music simultaneously, 8 to 10-year-old dyslexic children significantly improved both categorical perception of the VOT and their performance on reading, auditory memory and phonological tasks [210]. Although still preliminary, these results corroborate the intuitive use of activities such as music playing and dancing (among others) as authentic potential re-educational tools available to the practitioner to manage dyslexia and related disorders.

## 10. Conclusions

In conclusion to this vast albeit inevitably incomplete review of the literature on the brain mechanisms underpinning dyslexia and related disorders, two main trends seem to emerge as the most promising directions for future research as well as sound bases for constructing new therapies: altered long-range connectivity between several brain areas crucial for language as well as non-language functions in both hemispheres; and inaccurate processing of time-dependent information in its various dimensions, from the most elementary perceptual aspects to more sophisticated cognitive contents. Beyond the paradigmatic case of dyslexia, these principles could apply equally to a wider range of conditions known to be potentially associated with specific reading impairment.

In line with a Hebbian view of learning rules in neural systems, one might intuitively be inclined to suspect a close link between the two classes of data, but the amount of formal work done is still insufficient to document their reciprocal relationships, calling for further studies in neurotypical as well as dyslexic children. Interestingly enough, however, the current state of the art in these two research areas seems to converge toward a same, strong incitation to develop non-specific tools, such as the aforementioned musical or rhythmic ones, which, through more extensive use, could transfigure the field of therapy in neurodevelopmental disorders and hopefully also be applied in pedagogical practice.

## Figures and Tables

**Figure 1 brainsci-11-00708-f001:**
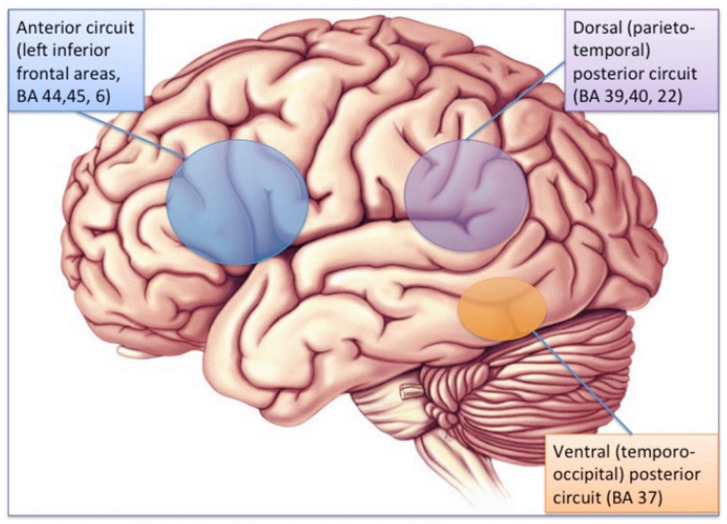
The brain reading network: left-hemisphere cortical regions showing consistent structural and functional abnormalities in phonological dyslexic adults and children [23].

**Figure 2 brainsci-11-00708-f002:**
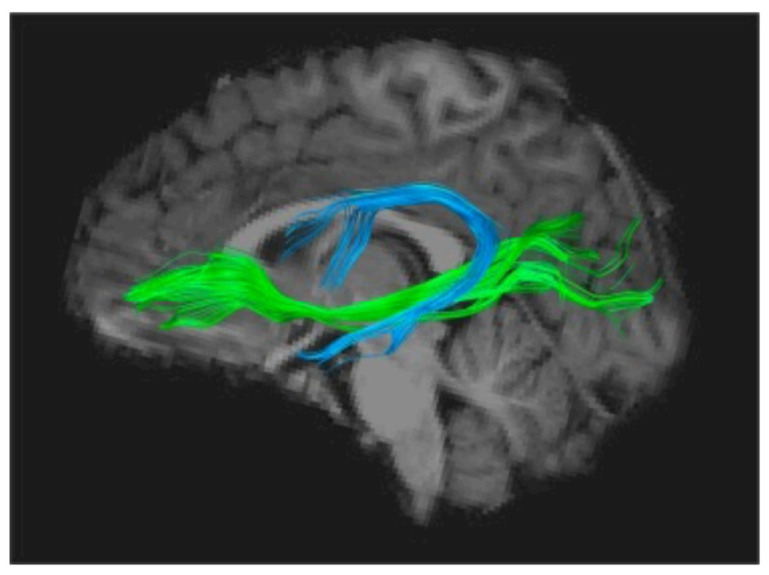
The two main white matter tracts usually reported as abnormally organized in the dyslexic brain: arcuate fasciculus (AF, in blue) and inferior fronto-occipital fasciculus (IFOF, in green), respectively associated with phonological and orthographic scores on standardized tests [30,33].

**Figure 3 brainsci-11-00708-f003:**
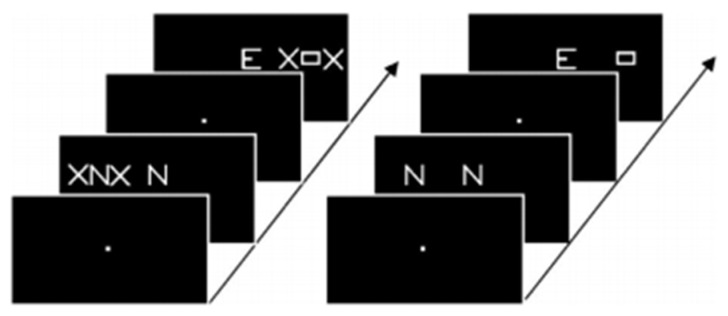
Example of stimuli used by Peyrin et al. [44] in their categorization task. Participants had to decide whether the stimuli of a pair were identical or not. Parafoveal stimuli were lateralized in either the right or left visual field, masked by two X in the flanked condition (**left side** of the figure) and displayed alone in the isolated condition (**right side**).

**Figure 4 brainsci-11-00708-f004:**
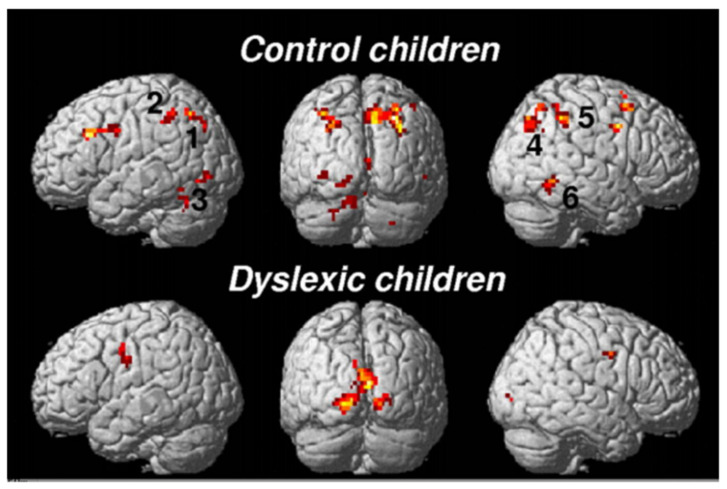
FMRI activations during the flanked condition of the categorization task in Peyrin et al. [44]: underactivation of the superior parietal cortex bilaterally in typical visuo-attentional dyslexics.

**Figure 5 brainsci-11-00708-f005:**
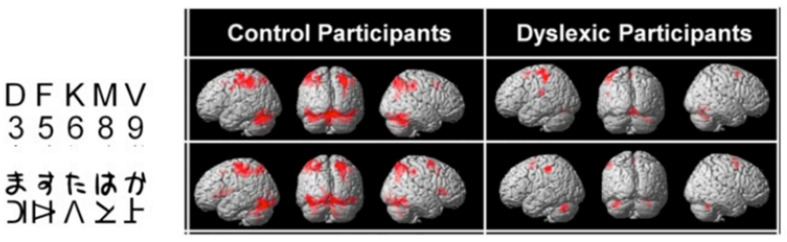
Brain activation by alphanumeric (**top**) and nonalphanumeric (**below**) strings in controls and dyslexics. Underactivation of the superior parietal lobules (mainly **right**) in dyslexics [45].

**Figure 6 brainsci-11-00708-f006:**
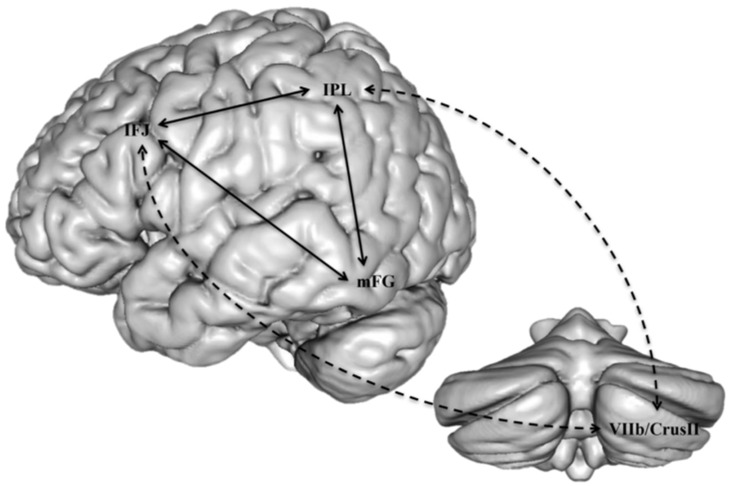
A presumed substrate of the modulatory role of the cerebellum on dorsal phonological processes in reading, indirectly affecting orthographic processes in the inferior temporal region (from [69]). IFJ: inferior frontal junction; IPL: inferior parietal lobule; mFG: mid-fusiform gyrus.

**Figure 7 brainsci-11-00708-f007:**
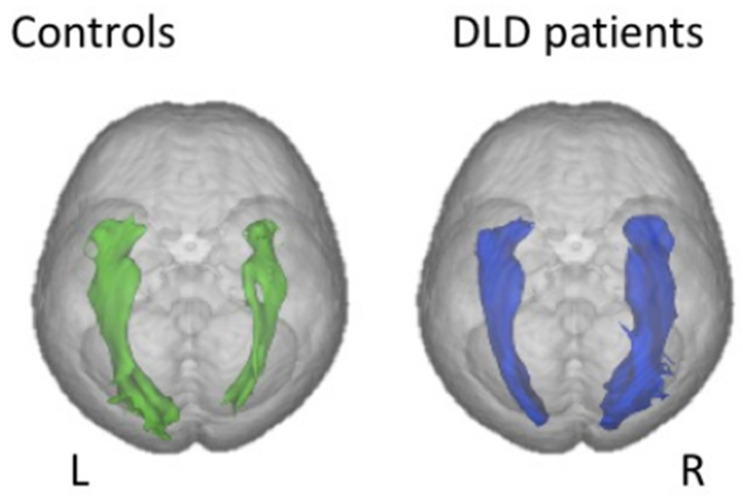
Diffusion MRI findings in children with developmental language disorder (right) compared to normal controls. Reversed rightward volume asymmetry in the IFOF (inferior fronto-occipital fasciculus). From [81].

**Figure 8 brainsci-11-00708-f008:**
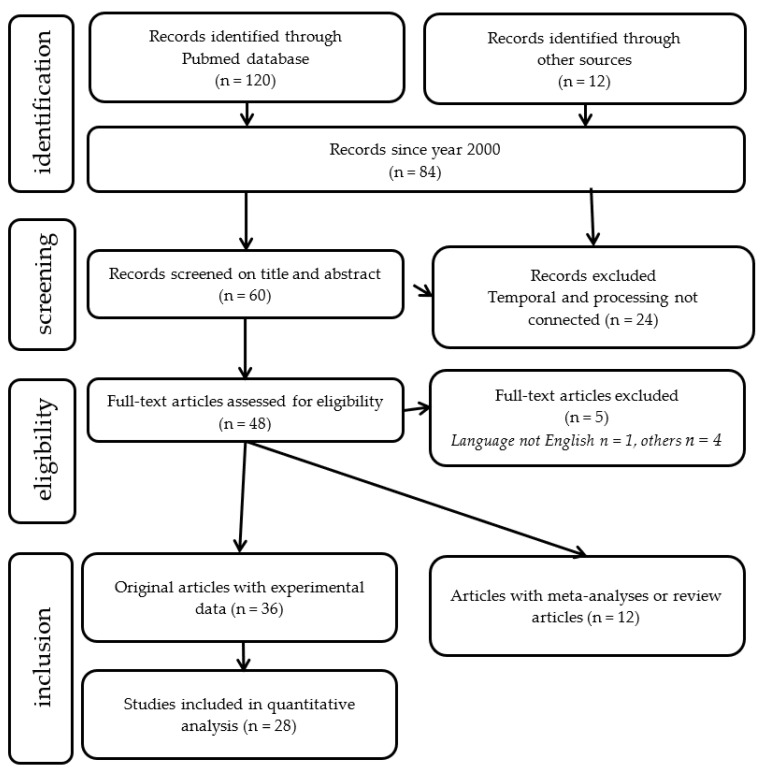
Search strategy: flowchart showing the number of included studies for the analytic and meta-analytic components of the review.

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
