# Peer review of "The Neurological Basis of Developmental Dyslexia and Related Disorders: A Reappraisal of the Temporal Hypothesis, Twenty Years on"

_brainsci, 2021, doi:10.3390/brainsci11060708_

Round 1

Reviewer 1 Report

This is an outstanding manuscript. I thoroughly enjoyed reading it, and have no recommendations. This will make a highly valuable contribution to the literature, as it brings together research on dyslexia and related neurodevelopmental disabilities in a unifying manner. The exquisite writing makes it accessible to a wide range of readers. 

Author Response

Reply to reviewer 1

This is an outstanding manuscript. I thoroughly enjoyed reading it, and have no recommendations. This will make a highly valuable contribution to the literature, as it brings together research on dyslexia and related neurodevelopmental disabilities in a unifying manner. The exquisite writing makes it accessible to a wide range of readers.

I really thank reviewer 1 for his/her very positive comment.

No response required

Reviewer 2 Report

This work offers an interesting and insightful review of the literature on the neurobiological bases of dyslexia and of comorbid disorders. However, as the author claims in the Conclusion, "this (is) inevitably incomplete and partial review of the literature on the brain mechanisms underpinning dyslexia and related disorders". Well, I would suggest the author to go beyond this limit and to apply the methodology required to produce a high-quality systematic review of the literature. The relevance of the topic would deserve this effort.

I guess the author made the selection of the literature according to his expertise and his personal evaluation of the relevance of the contributes, in order to offer a synthesis of the main results of the research in the field. However, if the author declared the inclusion criteria of the papers and the list of the sources from which the papers were selected, describing the flow chart of the selection process, the main results proposed by the manuscript would gain a higher degree of reliability. In other words, I invite the author to apply the methodology suggested by Cochrane guidelines to produce a high-quality systematic review on the neurobiological bases of dyslexia and on the efficacy of the interventions focused on the use of music and rhythm.

Author Response

" I would suggest the author to go beyond this limit and to apply the methodology required to produce a high-quality systematic review of the literature. The relevance of the topic would deserve this effort."

" I invite the author to apply the methodology suggested by Cochrane guidelines to produce a high-quality systematic review"

I have tried to follow the reviewer's advice in adding a new paragraph p. 18 entitled "Search strategy", where I describe step by step the process of literature search for the main topic of this review, i.e. the temporal processing hypothesis, introducing a flowchart provided in a new figure (fig. 8) which describes in details this process.

I have slightly modified the references list accordingly.

I hope this additional information will meet the requirements expressed by the reviewer.

Round 2

Reviewer 2 Report

The manuscript improved its scientific reliability by adding the research strategy on p. 18, so it might be considered fit for publication, in the present form.